# Association of Candidate Gene Polymorphism with Metabolic Syndrome among Mongolian Subjects: A Case-Control Study

**DOI:** 10.3390/medsci8030038

**Published:** 2020-09-02

**Authors:** Ariunbold Chuluun-Erdene, Orgil Sengeragchaa, Tsend-Ayush Altangerel, Purevjal Sanjmyatav, Batnaran Dagdan, Solongo Battulga, Lundiamaa Enkhbat, Nyamjav Byambasuren, Munkhzol Malchinkhuu, Munkhtstetseg Janlav

**Affiliations:** 1Department of Biochemistry, School of Biomedicine, Mongolian National University of Medical Sciences, Ulaanbaatar 14210, Mongolia; ariunbold.md@gmail.com (A.C.-E.); Sengeragchaa.orgil@gmail.com (O.S.); tsend-ayush@mnums.edu.mn (T.-A.A.); purevjal.s@gmail.com (P.S.); batnarandagdan@gmail.com (B.D.); solongobattulga@gmail.com (S.B.); lundiae93@gmail.com (L.E.); bnyamjav0909@gmail.com (N.B.); 2Coronary Care Unit, Cardiovascular Center, The Shastin Central Hospital, Ulaanbaatar 16081, Mongolia; 3Department of Pathophysiology, School of Biomedicine, Mongolian National University of Medical Sciences, Ulaanbaatar 14210, Mongolia; munkhzol@mnums.edu.mn

**Keywords:** metabolic syndrome, obesity, adipokines, insulin resistance

## Abstract

Metabolic syndrome (MetS) is complex and determined by the interaction between genetic and environmental factors and their influence on obesity, insulin resistance, and related traits associated with diabetes and cardiovascular disease risk. Some dynamic markers, including adiponectin (*ADIPOQ*), brain-derived neurotrophic factor *(BDNF*), and lipoprotein lipase (*LPL*), are implicated in MetS; however, the influence of their genetic variants on MetS susceptibility varies in racial and ethnic groups. We investigated the association of single nucleotide polymorphism (SNP)-SNP interactions among nine SNPs in six genes with MetS’s genetic predisposition in Mongolian subjects. A total of 160 patients with MetS for the case group and 144 healthy individuals for the control group were selected to participate in this study. Regression analysis of individual SNPs showed that the *ADIPOQ* + 45GG (odds ratio (OR) = 2.09, *p* = 0.011) and P^+^P^+^ of *LPL* PvuII (OR = 2.10, *p* = 0.038) carriers had an increased risk of MetS. Conversely, G allele of *LPL* S447X (OR = 0.45, *p* = 0.036) and *PGC-1α* 482Ser (OR = 0.26, *p* = 0.001) allele were estimated as protective factors, respectively. Moreover, a haplotype containing the G-P^+^-G combination was related to MetS. Significant loci were also related to body mass index (BMI), systolic blood pressure (SBP), serum high-density lipoprotein cholesterol (HDL-C), triglyceride (TG), and fasting blood glucose (FBG), adipokines, and insulin as well as insulin resistance (*p* < 0.05). Our results confirm that *ADIPOQ* + 45T > G, *LPL* PvII, and *PGC-1α* Gly482Ser loci are associated with MetS in Mongolian subjects.

## 1. Introduction

Metabolic syndrome MetS is a substantial global public health problem and concern because of its high prevalence (20–25% of the world’s adult population) and linked to more severe pathologies [1]. Those with MetS are at a three-fold increased risk of developing cardiovascular diseases (CVDs) and five-fold increased risk of type 2 diabetes mellitus (T2DM) [2]. Since the precise definition of MetS differs in specific details issued by World Health Organization (WHO) in 1998 [3], The American Heart Association (AHA) in 2001 [4], the National Cholesterol Educational Program Expert Panel on Detection, Evaluation, and Treatment of High Blood Cholesterol in Adults (ATP III definition) in 2001 [5], the National Heart, Lung and Blood Institute (NHLBI) in 2005 [6], and International Diabetes Federation (IDF) in 2006 [7], it is generally agreed that MetS is the compounding of several risk factors including abdominal obesity, insulin resistance, hyperglycemia, hyperlipidemia, and hypertension [1]. According to the IDF criteria, by 2015, the prevalence of MetS was estimated to occur in 32.7% in the general Mongolian population: socio-environmental factors including moderate-to-high alcohol consumption in men and widowed status in women are significantly associated with MetS [8].

The first-line intervention for MetS is to mitigate the modifiable, underlying risk factors (obesity, physical inactivity, and atherogenic diet) through lifestyle changes [9]. Effective lifestyle modification decreases the risk of developing type 2 diabetes among high-risk individuals by approximately 30–70% [10]. Then, if absolute risk is high enough, consideration can be given to incorporating drug therapy according to the existing guideline [11]. These reveal that the MetS, contrary to advanced type 2 diabetes and cardiovascular diseases, likely is a reversible condition if addressed early on, and long-term engagement in lifestyle changes may result in its resolution.

On the other hand, an irreversible risk factor is genetic background. Genetic influences play an essential role in the development of MetS in multiple ways, however, the mechanisms involved have not yet been fully understood. MetS’s fundamental components (obesity [12], dyslipidemia [13], hyperglycemia [14], and high blood pressure (BP) [15]) have a genetic basis, for which candidate genes have been studied. A survey that comprised 44 single nucleotide polymorphism (SNP)s of 31 candidate genes related to the lipid metabolism among Japanese people with MetS found that - 3A < G and 553G < T (Gly185Cys) polymorphisms of Apolipoprotein A5 (*APOA5*), the 2052T < C (Val653Val) and 1866C < T (Asn591Asn) polymorphisms of Low-density lipoprotein receptor (*LDLR*), the 13989ARG (Ile118Val) polymorphism of CYP3A*4* (*Cytochrome P450 3A4*) and the 1014T < A polymorphism of C1q and tumor necrosis factor-related protein 5 (*C1QTNF5*) were significantly associated with the prevalence of MetS [16]. Some researchers suggested that population-specific SNP, related to body mass index (BMI), waist circumference (WC), or energy metabolism, are probably inherited among Mongolians as a positive selection [17]. Therefore, to determine the genetic risk among Mongolians, it is necessary to study the common gene polymorphisms associated with MetS.

To our knowledge, these gene polymorphisms have not been determined in the Mongolian subjects, and our study is the first attempt to investigate their relationships between gene polymorphisms and MetS in Mongolian subjects. The current study was intended to determine the genetic effect of some candidate gene polymorphisms on features of MetS in Mongolian subjects.

## 2. Materials and Methods

### 2.1. Study Subjects

All participants signed written informed consent. The protocol was according to the Helsinki Declaration, and ethical approval was obtained from the Ethics Committee at Mongolian National University of Medical Science (MNUMS; protocol #201702/13-12/1A). A total of 160 MetS patients (86 males, 74 females; aged: 18–60) were selected from the Ulaanbaatar population for study. Inclusion criteria were based on a modified or harmonizing criterion as proposed in 2009, by the IDF and the AHA-NHLBI [18] that incorporates ethnicity by providing different criteria for MetS in different ethnic groups; Asian identifies for subjects with at least three of the criteria listed: abdominal obesity with WC ≥ 90 cm for men and ≥ 80 cm for women, systolic blood pressure (SBP) ≥ 130 mmHg, diastolic blood pressure (DBP) ≥ 85 mmHg, serum triglyceride (TG) level ≥ 150 mg/dL, serum high-density lipoprotein cholesterol (HDL-C) < 40/50 mg/dL for men/women, fasting blood glucose (FBG) ≥ 100 mg/dL. The presence of subjects with independent diseases, including coronary heart disease, diabetes mellitus, or chronic diseases (hypertension, hyperlipidemia, and dyslipidemia), were excluded from the case group. The control group consisted of 144 individuals (71 males, 73 females; aged: 18–60) with no history of obesity, hyperlipidemia, dyslipidemia, hypertension, or diabetes mellitus and confirmed by a health examination.

### 2.2. Biochemical Parameters

All blood samples were obtained from subjects after overnight fasting. Biochemical parameters were analyzed using commercial kits (AGAPPE DIAGNOSTICS SWITZERLAND GmbH, Knonauerstrasse 54-6330, Cham, Switzerland) for total cholesterol (TC), TG, HDL-C, and FBG. Low-density lipoprotein cholesterol (LDL-C) was calculated by the Friedewald formula. Homeostatic Model Assessment of Insulin Resistance (HOMA-IR) levels were measured by standard calculating methods: HOMA-IR = glucose (mg/dL) × insulin (μU/mL)/405. Serum adiponectin, leptin, and insulin levels were measured using a commercial direct enzyme-linked immunosorbent assay (ELISA) Human Adiponectin, Leptin, and Insulin kits, respectively, according to the procedure provided by the manufacturer (Linco Research, Inc., St. Louis, MO, USA).

### 2.3. Genotyping of SNPs

A total of nine SNPs: rs266729, rs2241766 of Adiponectin (*ADIPOQ*), rs6265 of Brain-derived neurotrophic factor *(BDNF*), rs688, rs5925 of Low-density lipoprotein receptor (*LDLR*), rs1805094 of Leptin receptor (*LEPR*)*,* rs285 and rs328 of Lipoprotein lipase (*LPL*), rs8192678 of Peroxisome proliferator-activated receptor gamma coactivator 1-alpha (*PGC-1α*) were selected as our study targets. Genomic DNA was isolated by a commercial kit, the “G-spin™ Total DNA Extraction Kit” (iNtRON Biotechnology, Seongnam, South Korea). According to previously published protocols [19,20,21,22,23,24,25,26], all SNPs were genotyped by a polymerase chain reaction and restriction fragment length polymorphism (PCR-RFLP) using Maxine PCR PreMix Kit (i-Star Taq; iNtRON Biotechnology, Seongnam, South Korea). The list of primers used in this study was summarized in Table 1.

### 2.4. Statistical Analysis

The quantitative data were represented as mean ± standard deviation (SD) or median and interquartile range (IQR). Statistical significance was evaluated by a *t*-test to compare differences between the two groups after skewed distributed values were normalized by natural logarithmic transformation. Qualitative data were shown as a percentage, further analyzed by Chi-square (χ^2^) test. The genotype distribution was compared between case and control groups with the χ^2^ test (3 × 2). The allele frequency was determined using direct gene counting analysis and χ^2^ test (2 × 2). For assessing the effect of the SNP genotype on the development of MetS, a multiple logistic regression analysis was performed with a 95% confidence interval (CI). In terms of the association of appreciable polymorphisms with clinical features, each polymorphism’s genotype was transformed into a genetic model that constituted of two groups: a dominant group of wild-type homozygotes versus heterozygotes and homozygotes and the recessive group consisting of wild-type homozygotes and heterozygotes versus homozygotes. General data analyses of the case-control study were conducted using SPSS 21.0 (IBM corporation, Chicago, IL, USA). Haplotype and pairwise linkage disequilibrium (LD) analysis were carried out by SNPStats online software (Institut Català d’Oncologia, Barcelona, Spain; https://www.snpstats.net/start.htm).

## 3. Results

### 3.1. Clinical Data and Biochemical Parameters

The baseline profile of MetS patients and controls is summarized in Table 2. The mean age of patients with MetS was 41.7 ± 11.3 and the control group was 41.2 ± 10.2. The proportion of BMI ≥ 25 kg/m^2^ among MetS and the control group was 64.1% and 35.9%. Of the individuals with WC ≥ 80 cm, 60.3% had MetS and 39.7% were in the control group. SBP and DBP levels were higher in patients with MetS than the control group (*p* < 0.001). Serum concentrations of FBG and TG were higher in the MetS group than the control group (*p* = 0.012, *p* < 0.001). In contrast, HDL-C was lower (*p* = 0.048) in the MetS group than the control group. Our data illustrated that insulin, HOMA-IR and leptin levels were also higher in the MetS group (*p* < 0.001).

### 3.2. Allele Frequency of SNPs

Genotype frequencies of nine SNPs in the MetS group and control group were calculated (Table 3). The χ^2^ test revealed that four of the identified SNPs were significantly related to the prevalence of MetS.

Results of the multiple logistic regression model, used to determine the significance of alleles of the two SNPs as a probable independent risk factor for MetS, demonstrated that heterozygous (TG + GG) of *ADIPOQ* + 45T > G (odds ratio (OR) = 1.98; 95%CI, 1.14–3.44; *p* = 0.015), and homozygous (P^+^P^+^) of *LPL* PvuII (OR = 2.10; 95%CI, 1.04–4.26; *p* = 0.038) carriers had an increased risk for development of MetS compared with those wild-type or the most frequent genotypes (Table 4). On the other hand, heterozygotes (CG + GG) of *LPL* S447X (OR = 0.45; 95%CI, 0.21–0.95; *p* = 0.036) and homozygotes (SS) of *PGC-1α* Gly482Ser (OR = 0.26; 95%CI, 0.12–0.58; *p* = 0.001) were independently estimated as a protective factor for MetS.

### 3.3. SNP-SNP Interaction and MetS

We analyzed the pairwise linkage disequilibrium (LD) pattern for significant polymorphisms: *ADIPOQ* + 45T > G (rs2241766), *LPL* PvuII (rs285), and *PGC-1α* Gly482Ser (rs8192678) (Figure 1). No SNP combination was in strong LD (D’ > 0.7, r^2^ > 0.25). Furthermore, we carried out haplotype analysis using multiple logistic regression models and listed all combinations accounted for more than 5% of the frequency (Table 5). Out of possible variants, we found (G-P^+^-G) haplotype significantly associated with the development of MetS (OR = 3.28; 95%CI, 1.32–8.16; *p* = 0.011).

### 3.4. Association of SNPs with Clinical Parameters

Clinical characteristics of all enrolled people were compared according to the dominant and recessive genotype of the SNPs relevant to MetS (Table 6). Our results suggest that the HDL-C and adiponectin levels were reduced more in *ADIPOQ* + 45T > G carriers (TG + GG) than those non-carriers (TT) in the dominant model (*p* = 0.032, *p* = 0.027, respectively). In the recessive, we found that BMI and blood pressure levels were higher in the homozygous carrier group (GG; *p* = 0.012, *p* = 0.001) of the SNP.

For *LPL* PvuII, (P^+^P^+^) genotype carriers had elevated TG and FBG levels compared with combined genotypes (P^+^P^-^ + P^-^P^-^; *p* = 0.009, *p* < 0.001). Meanwhile, (P^+^) allele (P^+^P^+^ + P^+^P^-^) in the recessive model showed elevated BMI and insulin levels than compared to non-carrier homozygotes (P^-^P^-^; *p* = 0.027, *p* = 0.027). Whereas (P^+^) allele carriers seemingly had more leptin secretion in both models (*p* = 0.001, *p* < 0.001).

With regard to the *PGC-1α* Gly482Ser, the recessive (SS) genotype carriers had elevated levels of SBP than (G) allele carriers (*p* = 0.004, *p* = 0.012) but our results disclosed higher levels of HOMA-IR in (GG + GS) genotype carriers which are opposite of levels found in (SS) genotype carriers (*p* = 0.038). Moreover, (GG) homozygotes were significantly associated with low levels of HDL-C (*p* = 0.011).

## 4. Discussion

The estimation of environmental and genetic influences on each MetS component may vary in different populations and ethnicities. Some genome-wide prediction surveys focused on associations with the MetS or clinical criteria [27,28]. Whereas, other studies of population-specific SNPs that were selected for their potential contribution to carbohydrate and lipid metabolism found that some gene polymorphisms were strongly related to FBG and TG levels [29,30]. We have investigated the gene variants affecting dynamic markers: *BDNF*, *LEPR*, *VLDLR,* and *PGC-1α*, but included loci related to cellular lipid metabolism, *LPL* and *ADIPOQ* were also identified in Mongolian patients with MetS. Only four out of nine tested gene variants, the *ADIPOQ* + 45T > G, *LPL* PvuII and S447X, *PGC-1α* Gly482Ser, showed a nominal difference in the initial genotype comparison.

### 4.1. Adiponectin (ADIPOQ)

The effects of genetic polymorphisms of *ADIPOQ* on the risk of obesity, T2D, and hypertension incidence have been studied in several ethnic groups. Among the Chinese Han population, *ADIPOQ* + 45T > G has been determined to be related to MetS [31]. Similarly, we have now shown that the ADIPOQ + 45T > G is associated with MetS. Moreover, we found significant associations between the polymorphism of the *ADIPOQ* and BMI, SBP, and HDL-C levels in the genetic models apart from the adiponectin level (Table 6). Our results are in agreement with other findings from the meta-analysis surveys that indicate *ADIPOQ* + 45T > G polymorphism is associated with hypertension and dyslipidemia phenotypes [31]. A silent mutation, from the changing substitution of (T) with (G) in the + 45T > G polymorphism that occurs in exon 2, may affect transcription rate, splicing, and mRNA transport regulation, probably explain this association. A haplotype analysis, based on a dense SNP map in a large sample of non-Hispanic whites and African Americans, clarified gene interaction involving haplotypes in the promoter and coding region, a 2-block linkage disequilibrium structure of the *ADIPOQ* impacting the level of plasma adiponectin [32]. From the survey, blocks have at least one SNP related to serum adiponectin levels. The haplotypes in the first block were linked to increased adiponectin levels. In contrast, the haplotypes in the second one were related to decreased adiponectin levels. Hence, the + 45T > G polymorphism is significantly associated with reduced serum adiponectin levels and high blood pressure. However, the mechanism of whether the + 45T > G polymorphism influences the hypertension susceptibility through the low levels of plasma adiponectin warrants further study. Another study, inversely, did not detect such effects or provided contradictory results concerning the polymorphic sites that are involved with an alteration in blood lipids [33] and glucose [34]. Nevertheless, it should be noted that several recent genome-wide association studies (GWAS) have been carried out on MetS components: obesity, dyslipidemia, high BP and T2D individually in several populations, most of which were conducted in people of European descent. The interactions between genetic factors, such as SNPs in the Adiponectin itself, and environmental factors causing obesity, may play a crucial role in developing insulin resistance, type 2 diabetes, and the MetS.

### 4.2. Lipoprotein Lipase (LPL)

The PvuII and S447X polymorphisms are located on intron 1 and exon 9, respectively. The genetic association has been described for PvuII and S447X polymorphisms in the *LPL* gene with various pathological conditions, including dyslipidemia, hypertension, CVD, and T2D. The meta-analysis study revealed a significant protective association between Ser447Ter and PvuII and the stroke risk [35]. Our findings indicated that PvuII is a risk factor for MetS, while the S447X has an inverse effect. In terms of PvuII, we also found increased levels of BMI and FBG. A similar tendency was reported by Bozina et al. [36], where they indicated that the PvuII (P^+^P^+^) genotype carrier had a higher level of BMI and glucose. Besides, the combination of two mutant alleles in HindIII (rs320) and PvuII was related to the increased level of WC [37] a potential indicator of visceral fat accumulation, which is closely associated with obesity-related MetS. Interestingly, increased levels of TG, leptin and insulin were recorded in PvuII (P^+^) allele carriers in our study. This coincidence might depict a dysregulation of energy homeostasis in coordinated leptin and insulin resistance due to elevated TG levels. A recent report indicates that an inhibition of leptin receptor by the serum TG in the brain induces central resistance to leptin and insulin for their centrally mediated effect on body weight [38], which probably is consistent with our findings. Concerning the S447X polymorphism, the only case of (GG) genotype was observed in the MetS group in the current study, except for combination genotype frequency. Due to that, we did not analyze it with biochemical parameters.

### 4.3. Peroxisome Proliferator-Activated Receptor Gamma Coactivator 1-Alpha (PPARGC1α or PGC-1α) 

Combining the initial and the replication analysis presented a 0.26–0.42-fold decrease in MetS risk associated with the (S) allele in our study. At variance, this result is the only negative study published thus far [39], in which no association was reported in Danish subjects with MetS. We found a significant association between *PGC-1α* Gly482Ser polymorphism and insulin resistance or HOMA-IR and SBP as well as HDL-C in the genetic model analysis (Table 4). Interestingly, our study indicated that the (S) allele was estimated as a protector factor for MetS, while the (S) allele carriers are more insulin resistant. Our results are consistent with the previous study reporting an association with insulin resistance [40]. However, we should point out a possible explanation for this issue because patients with the MetS may already be the result of insulin resistance, making it more complicated to distinguish between subjects within the MetS group. *PGC*-*1α* enhances the activity of other PPARs that bind to sequence-specific target elements in the promoter region of target genes that affect metabolic pathways such as insulin-regulated gluconeogenesis, glucose uptake by muscle cells [41]. Thus, a reduction in *PGC-1α* activity is probably caused by the transcriptional effects of Gly482Ser mutation, which might impact the metabolic pathways related to T2DM. To the association of the *PGC-1α* polymorphisms with hypertension, it was reported that subjects with (SS) homozygote genotype have lower SBP, which is similar to our findings, and these subjects have a much lower risk of developing hypertension than with (SS) genotype in Danish subjects [42]. Several mechanisms have been proposed for the antihypertensive effects of the *PGC-1α*. The *PGC-1α* loss of function was associated with a reduction in endothelial nitric oxide synthase (eNOS) expression in the endothelium. Conversely, a *PGC-1α* gain of function increased basal eNOS expression [43]. The findings of this study suggested that endothelial *PGC-1α* protects from vascular dysfunction.

In summary, our findings indicate that *ADIPOQ*, *LPL* and *PGC-1α* gene polymorphisms can determine genetic susceptibility to MetS as individual biomarkers and their synergistic interaction. On the other hand, genetic studies have provided only limited evidence for a common genetic background of the MetS. Epigenetic factors (DNA methylation and histone modification) are likely to play essential roles in the MetS. Extensive research is needed to clarify the role of genetic variation and epigenetic molecular mechanisms in MetS. We acknowledge the following limitations in the current study. First, selection bias was inevitable; participants of 18–60 age were selected, which might lead to the non-normal distribution of some biochemical parameters. However, we controlled for this by adjusting results with age. Second, our findings are based on a small number of case subjects and need to be replicated in a large population. Third, SNP–environment interactions should also be evaluated to account for MetS risk comprehensively. Nevertheless, our study supports the findings that affecting carbohydrate and lipid metabolism; gene polymorphisms are independent risk factors for the development of MetS.

## 5. Conclusions

Our results suggest that *ADIPOQ* + 45T > G, *LPL* PvII and *PGC-1α* Gly482Ser loci may contribute to the risk development of MetS in Mongolian subjects.

## Figures and Tables

**Figure 1 medsci-08-00038-f001:**
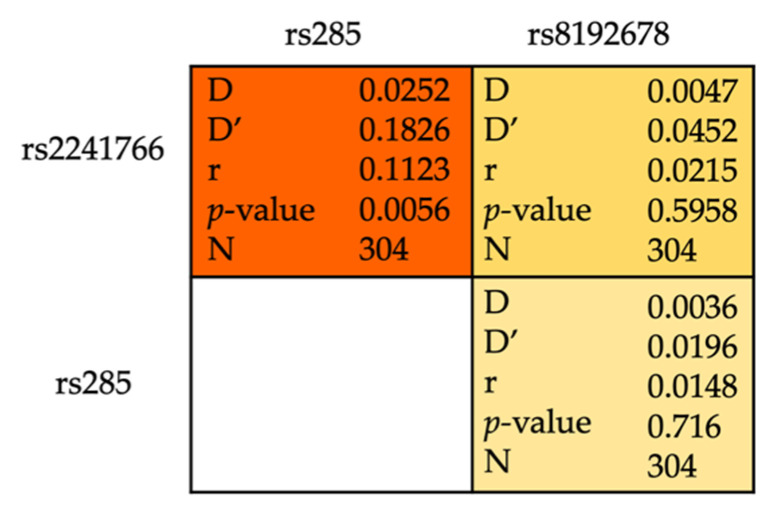
Paired single nucleotide polymorphism (SNP) linkage disequilibrium analysis.

**Table 1 medsci-08-00038-t001:** Primers and restriction enzymes.

Genes	SNPs	Primers	Restriction Enzymes
*ADIPOQ*	- 11377C > G (rs266729)	F: 5′-ACTTGCCCTGCCTCTGTCTG-3′ R: 5′-CCTGGAGAACTGGAAGCTG-3′	*HhaI*
	+ 45T > G (rs2241766)	F: 5′-GAAGTAGACTCTGCTGAGATGG-3′ R: 5′-TATCAGTGTAGGAGGTCTGTGATG-3′	*SmaI*
*BDNF*	Val66Met (rs6265)	F: 5′-ATCCGAGGACAAGGTGGC-3′ R: 5′-CCTCATGGACATGTTTGCAG-3′	*Eco72I*
*LDLR*	C1773T (rs688)	F: 5’-TCTCCTTATCCACTTGTGTGT-3′ R: 5’-CTTCGATCTCGTACGTAAGC-3′	*HincII*
	AvaII (rs5925)	F: 5′-GTCATCTTCCTTGCTGCCTGTTTAG-3′ R: 5′-GTTTCCACAAGGAGGTTTCAAGGTT-3′	*AvaII*
*LEPR*	K656N (rs1805094)	F: 5′-ACTAGATGGACTGGGATATTGGAGTAAT-3′ R: 5′-CTTCCAAAGTAAAGTGACATTTTTCGC -3′	*BstUI*
*LPL*	PvuII (rs285)	F: 5′-ATCAGGCAATGCGTATGAGGTAA-3′ R: 5′-GAGACACAGATCTCTTAAGAC-3′	*PvuII*
	S447X (rs328)	F: 5′-TACACTAGCAATGTCTAGGTGA-3′ R: 5′-TCAGCTTTAGCCCAGAATGC-3′	*MnlI*
*PGC-1α*	Gly482Ser (rs8192678)	F: 5′-GAAGTCCTCAGTCCTCAC-3′ R: 5′-GGGGTCTTTGAGAAAATAAGG-3′	*MspI*

SNP, single nucleotide polymorphism; rs, reference SNP cluster id; *ADIPOQ*, Adiponectin; *BDNF*, Brain-derived neurotrophic factor; *LDLR*, Low-density lipoprotein receptor; *LEPR*, Leptin receptor; *LPL*, Lipoprotein lipase; *PGC-1α*, Peroxisome proliferator-activated receptor gamma coactivator 1-alpha.

**Table 2 medsci-08-00038-t002:** Main characteristics of the MetS and control group.

Parameters	MetS	Control	*p*-Value
Age, yr	41.7 ± 11.3	41.2 ± 10.2	0.924
Gender (M/F)	160 (86/74)	144 (71/73)	0.527
BMI, kg/m^2^	31.27 ± 4.23	26.64 ± 3.75	<0.001
WC, cm	100.97 ± 1.10	89.01 ± 12.75	<0.001
SBP, mmHg	128.75 ± 13.97	114.02 ± 14.44	<0.001
DBP, mmHg	88.42 ± 9.92	77.95 ± 9.52	<0.001
FBG, mg/dL	92.07 ± 66.57	71.69 ± 12.69	0.012
TC, mg/dL	157.72 ± 36.42	148.47 ± 36.73	0.123
TG, mg/dL ^1^	124.45 (83.74–179.37)	66.37 (49.52–96.35)	<0.001
HDL-C, mg/dL	32.04 ± 11.49	36.44 ± 15.63	0.048
LDL-C, mg/dL	96.85 ± 40.38	95.75 ± 39.53	0.867
Adiponectin, ng/mL ^1^	6.46 (0.06–19.51)	6.18 (0.09–49.66)	0.082
Leptin, ng/mL ^1^	11.10 (2.30–56.30)	4.5 (0.01–34.87)	<0.001
Insulin, mIU/mL ^1^	12.06 (0.29–113.53)	8.53 (0.29–170.29)	<0.001
HOMA-IR ^1^	2.28 (0.05–38.99)	1.43 (0.05–36.32)	<0.001

MetS, metabolic syndrome; BMI, body mass index; M, male; F, female; WC, waist circumference; SBP, systolic blood pressure; DBP, diastolic blood pressure; FBG, fasting blood glucose; TG, triglyceride; TC, total cholesterol; HDL-C, high-density lipoprotein cholesterol; LDL-C, low-density lipoprotein cholesterol; HOMA-IR, homeostatic model assessment-insulin resistance. Values for continuous variables are expressed as the mean ± standard deviation (SD) and the median and interquartile range (IQR). ^1^ After log-transformed, the *t*-test was utilized for comparison.

**Table 3 medsci-08-00038-t003:** Genotype frequencies of SNPs.

Genes	SNPs	Genotype	MetS *n* (%)	Control Group *n* (%)	*p*-Value
*ADIPOQ*	- 11377C > G	CC	82 (51.3)	71 (49.3)	0.775
	(rs266729)	CG	70 (43.6)	63 (43.7)	
		GG	8 (5.1)	10 (7.0)	
	+ 45T > G	TT	72 (45.0)	85 (59.0)	0.002
	(rs2241766)	TG	80 (50.0)	44 (30.5)	
		GG	8 (5.0)	15 (10.4)	
*BDNF*	Val66Met	Met/Met	33 (21.2)	54 (37.5)	0.157
	(rs6265)	Val/Met	119 (73.8)	84 (58.3)	
		Val/Val	8 (5)	6 (4.2)	
*LDLR*	C1773T	CC	2 (1.2)	4 (2.8)	0.578
	(rs688)	CT	97 (60.9)	66 (45.7)	
		TT	61 (37.8)	74 (51.4)	
	AvaII	A^-^A^-^	90 (56.2)	71 (49.3)	0.976
	(rs5925)	A^-^A^+^	70 (43.8)	65 (45.1)	
		A^+^A^+^	-	8 (5.6)	
*LEPR*	K656N	GG	128 (76.3)	110 (76.4)	0.865
	(rs1805094)	GC	26 (16.3)	26 (18.1)	
		CC	6 (7.5)	8 (5.6)	
*LPL*	PvuII	P^-^P^-^	51 (31.8)	40 (27.8)	0.028
	(rs285)	P^-^P^+^	50 (31.3)	66 (45.8)	
		P^+^P^+^	59 (36.9)	38 (26.4)	
	S447X	CC	143 (89.3)	106 (73.6)	<0.001
	(rs328)	CG	16 (10.0)	33 (22.9)	
		GG	1 (6.2)	5 (3.4)	
*PGC-1α*	Gly482Ser	GG	83 (51.8)	48 (33.4)	0.004
	(rs8192678)	GS	55 (34.3)	69 (47.9)	
		SS	22 (13.7)	27 (18.7)	

SNPs, single nucleotide polymorphisms; MetS, metabolic syndrome. Genotype distributions of each SNP were compared between case and control groups using the χ^2^ test (3 × 2).

**Table 4 medsci-08-00038-t004:** Genotype frequencies of polymorphisms associated with MetS.

Gene/SNP	Genotype	MetS *n* (%)	Control Group *n* (%)	OR * (95%CI)	*p*-Value
*ADIPOQ* + 45T > G (rs2241766)	TT	72 (45.0)	85 (59.0)	1.00	
	TG	80 (50.0)	44 (30.5)	1.39 (0.44–3.18)	0.570
	GG	8 (5.0)	15 (10.4)	2.09 (1.18–3.72)	0.011
	TT/TG + GG	88 (55.0)	59 (41.0)	1.98 (1.14–3.44)	0.015
	TT + TG/GG	8 (5.0)	15 (10.4)	0.53 (0.18–1.10)	0.081
*LPL* PvuII (rs285)	P^-^P^-^	51 (31.8)	40 (27.8)	1.00	
	P^-^P^+^	50 (31.3)	66 (45.8)	0.85 (0.43–1.69)	0.661
	P^+^P^+^	59 (36.9)	38 (26.4)	2.10 (1.04–4.26)	0.038
	P^-^P^-^/P^-^P^+^ + P^+^P^+^	109 (68.1)	104 (72.2)	1.31 (0.72–2.39)	0.370
	P^-^P^-^ + P^-^P^+^/P^+^P^+^	59 (36.9)	38 (26.4)	2.29 (1.26–4.18)	0.006
*LPL* S447X (rs328)	CC	143 (89.3)	106 (73.6)	1.00	
	CG	16 (10.0)	33 (22.9)	0.17 (0.02–1.54)	0.117
	GG	1 (6.2)	5 (3.4)	0.52 (0.23–1.15)	0.106
	CC/CG + GG	17 (10.6)	38 (26.4)	0.45 (0.21–0.95)	0.036
	CC + CG/GG	1 (6.2)	5 (3.4)	0.19 (0.02–1.70)	0.139
*PGC1* Gly482Ser (rs8192678)	GG	83 (51.8)	48 (33.4)	1.00	
	GS	55 (34.3)	69 (47.9)	0.41 (0.22–0.77)	0.006
	SS	22 (13.7)	27 (18.7)	0.26 (0.12–0.58)	0.001
	GG/GS + SS	77 (48.1)	96 (66.7)	0.36 (0.20–0.63)	<0.001
	GG + GS/SS	22 (13.7)	27 (18.7)	0.42 (0.20–0.83)	0.013

MetS, metabolic syndrome; SNP, single nucleotide polymorphism; OR, odds ratio; CI, confidence interval. *Adjusted with age, gender, BMI and WC.

**Table 5 medsci-08-00038-t005:** Association of Haplotype with MetS risk.

	rs2241766	rs285	rs8192678	Frequency		OR * (95%CI)	*p*-value
				MetS	Control		
1	T	P^-^	G	0.264	0.249	1.00	-
2	T	P^+^	G	0.204	0.168	0.83 (0.40–1.73)	0.630
3	T	P^+^	S	0.124	0.182	1.27 (0.65–2.51)	0.49
4	G	P^+^	G	0.152	0.100	3.28 (1.32–8.16)	0.011
5	G	P^-^	G	0.066	0.05	1.39 (0.30–6.48)	0.683
6	G	P^-^	S	0.044	0.063	1.58 (0.49–5.07)	0.440
7	G	P^+^	S	0.037	0.042	1.26 (0.24–6.65)	0.788

MetS, metabolic syndrome; OR, odds ratio; CI, confidence interval. * Adjusted with age, gender, BMI and WC.

**Table 6 medsci-08-00038-t006:** Clinical features of subjects according to the genotype of polymorphisms.

SNP	Parameters	Dominant Model ^a^			Recessive Model ^b^		
		AA	Ab + bb	*p*-Value	AA + Ab	bb	*p*-value
*ADIPOQ* + 45T > G (rs2241766)	BMI, kg/m^2^	28.92 ± 4.41	29.32 ± 4.83	0.450	26.69 ± 5.29	29.29 ± 4.52	0.012
	WC, cm	90.57 ± 9.13	95.80 ± 13.48	0.082	94.82 ± 13.51	96.08 ± 13.03	0.412
	SBP, mmHg	121.59 ± 17.18	122.58 ± 14.55	0.597	111.42 ± 8.38	122.91 ± 16.06	0.001
	TC, mg/dL	149.76 ± 37.80	156.62 ± 35.40	0.105	152.50 ± 32.58	153.37 ± 37.05	0.913
	TG, mg/dL ^1^	82.74 (55.93–149.90)	94.35 (66.34–139.40)	0.629	90.09 (66.68–169.40)	93.40 (56.64–143.40)	0.845
	HDL-C, mg/dL	35.66 ± 14.47	32.40 ± 12.62	0.032	33.68± 13.00	38.78±14.85	0.086
	LDL-C, mg/dL	97.04 ± 40.96	95.44 ± 38.52	0.730	96.62 ± 40.00	92.03 ±36.98	0.595
	FBG, mg/dL	77.68 ± 39.12	87.47 ± 59.01	0.088	81.87±50.15	89.06±47.26	0.508
	Adiponectin, ng/mL ^1^	6.41 (4.05–10.38)	6.33 (2.82–10.18)	0.027	6.51 (4.02–10.31)	5.28 (3.42–12.91)	0.660
	Leptin, ng/mL ^1^	7.5 (4.37–15.75)	7.2 (3.85–19.87)	0.559	7.20 (4.34–16.46)	7.41 (2.12–18.74)	0.127
	Insulin, mIU/mL ^1^	10.88 (5.88–15.59)	11.18 (5.59–17.06)	0.736	11.18 (5.59–16.62)	9.12 (5.29–13.82)	0.256
	HOMA-IR ^1^	1.87 (0.97–3.10)	1.84 (1.04–3.57)	0.828	1.87 (1.03–3.22)	1.81 (0.98–3.04)	0.468
*LPL* PvuII (rs285)	BMI, kg/m^2^	29.42 ± 4.89	28.45 ± 3.91	0.088	30.00 ± 4.83	28.72 ± 4.47	0.027
	WC, cm	95.45 ± 11.26	95.42 ± 14.16	0.984	97.40 ± 13.53	94.55 ± 13.09	0.088
	SBP, mmHg	123.19 ± 17.88	121.56 ± 14.90	0.416	122.40 ± 16.39	121.36± 14.79	0.610
	TC, mg/dL	157.36 ± 34.00	151.47 ±37.76	0.197	152.22 ±36.40	155.78±37.39	0.438
	TG, mg/dL ^1^	130.75 (86.74–187.50)	117.30 (82.74–174.80)	0.009	113.60 (78.31–185.40)	150.80 (97.98–176.32)	0.787
	HDL-C, mg/dL	32.87 ± 14.10	34.65 ± 13.47	0.296	33.99 ± 13.32	34.27 ± 14.51	0.870
	LDL-C, mg/dL	97.37 ± 39.54	95.75 ± 39.91	0.745	99.01 ± 41.22	95.02 ± 39.08	0.425
	FBG, mg/dL	99.90 ± 79.70	74.34 ± 22.85	<0.001	85.01 ± 56.51	76.44 ± 28.89	0.082
	Adiponectin, ng/mL ^1^	5.60 (4.05–7.21)	7.23 (5.20–10.01)	0.078	6.23 (4.33–9.74)	7.11 (5.17–8.98)	0.057
	Leptin, ng/mL ^1^	14.65 (7.50–27.80)	9.60 (6.40–22.60)	0.001	15.95 (7.50–25.10)	7.80 (4.60–9.70)	<0.001
	Insulin, mIU/mL ^1^	13.97 (7.94–18.53)	11.76 (5.59–21.76)	0.113	12.06 (7.35–21.76)	11.47 (6.03–19.78)	0.021
	HOMA-IR ^1^	2.32 (1.19–4.06)	2.28 (1.23–4.26)	0.997	2.31 (1.64–4.37)	2.23 (1.11–4.07)	0.194
*PGC-1α* Gly482Ser (rs8192678)	BMI, kg/m^2^	29.13 ± 4.71	29.10 ± 4.55	0.958	29.00 ± 4.87	29.65 ± 2.99	0.368
	WC, cm	95.97 ± 14.29	95.01 ± 12.46	0.537	98.12 ± 10.76	94.89 ± 13.68	0.117
	SBP, mmHg	125.07 ± 16.37	119.75 ± 15.18	0.004	122.93 ± 16.59	118.00 ± 11.33	0.012
	TC, mg/dL	155.64 ± 37.50	150.22 ± 35.48	0.204	154.37 ± 35.98	147.97 ± 39.99	0.261
	TG, mg/dL ^1^	99.08 (56.36–169.25)	90.57 (59.83–129.20)	0.137	94.35 (56.92–147.40)	86.74 (59.60–121.55)	0.404
	HDL-C, mg/dL	31.76 ± 11.65	35.82 ± 14.82	0.011	33.48 ± 12.91	37.04 ± 16.83	0.092
	LDL-C, mg/dL	98.71 ± 41.50	93.05 ± 37.20	0.226	97.39 ± 39.06	90.73 ± 42.86	0.281
	FBG, mg/dL	84.46 ± 59.93	80.89 ± 40.96	0.538	82.60 ± 52.33	81.45 ± 35.46	0.881
	Adiponectin, ng/mL ^1^	6.28 (4.02–10.61)	6.68 (3.50–10.06)	0.087	6.33 (3.77–10.46)	6.95 (2.46–8.85)	0.314
	Leptin, ng/mL ^1^	7.20 (4.40–16.46)	7.01 (3.90–17.30)	0.210	7.00 (3.91–15.67)	8.51 (4.71–20.24)	0.519
	Insulin, mIU/mL ^1^	11.18 (6.39–14.48)	10.29 (5.29–17.06)	0.830	10.29 (5.59–14.41)	12.35 (7.86–21.03)	0.067
	HOMA-IR ^1^	1.76 (1.10–2.90)	1.95 (0.91–3.43)	0.142	1.81 (0.98–3.04)	2.52 (1.25–4.39)	0.038

Values for continuous variables are expressed as the mean ± standard deviation (SD) and as the median and interquartile range (IQR). ^1^ After log-transformed, the *t*-test was utilized for comparison. ^a^ Dominant model TT/TG + GG for *ADIPOQ* + 45T > G, P^+^P^+^/P^-^P^+^ + P^-^P^-^ for *LPL* PvuII, GG/GS + SS for *PGC-1α* Gly482Ser. ^b^ Recessive model TT + TG/GG for *ADIPOQ* + 45T > G, P^+^P^+^ + P^-^P^+^/P^-^P^-^ for *LPL* PvuII, GG + GS/SS for *PGC-1α* Gly482Ser.

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
