# Peer review of "Association of Candidate Gene Polymorphism with Metabolic Syndrome among Mongolian Subjects: A Case-Control Study"

_medsci, 2020, doi:10.3390/medsci8030038_

Round 1
Reviewer 1 Report
An excellent paper that took multiple factors into consideration when selecting participants and analyzing data. My only additional consideration would be to include a diagramatic representation of the interactions of the SNP polymorphisms and implications for the development of MS. It would also be interesting to note how many of the control subjects have a familial history of MS and if those subjects have any of the SNP polymorphisms.
Author Response
Dear reviewer,
Thank you for your review. All revisions we made were highlighted by “track changes” in the manuscript.
We made the following revision;
1. For the interaction of single nucleotide polymorphisms (SNPs) we chose, we tried to find out new results that can show it with SNP analysis. We first constructed Linkage Disequilibrium Block for interaction between three significant SNPs as Figure 1 in line 279. Unfortunately, we did not find strong relation/interaction among them; however, we performed haplotype analysis regarding SNP-SNP interaction to imply the MS risk. Luckily, we find a significant result from it and then presented as Table 5 in the manuscript.
2. We agree with you that taking an accurate familial history of MS is invaluable when considering an inheritance for their offspring. Since our raw data did not obtain familial history questionnaires from study participants, we apologize that it would be impossible to show. As you can see, genotype and haplotype frequency illustrate how many of the control subjects have any significant SNP polymorphisms, despite the familial history.
We welcome any comments. Thank you.
Reviewer 2 Report
The manuscript titled “Association of Candidate Gene Polymorphism with Metabolic Syndrome among Mongolian Subjects: A Case-Control Study” by Chuluun-Erdene et al identifies a role for the gene polymorphism in the patients of metabolic syndrome among Mongolian subjects. In detail, the authors specifically, they try to link the relationship in the polymorphisms of Adiponectin (ADIPOQ), Brain-derived neurotrophic factor (BDNF), Low-density lipoprotein receptor (LDLR), Leptin receptor (LEPR), Lipoprotein lipase (LPL) and Peroxisome proliferator-activated receptor gamma coactivator 1-alpha (PGC-1α) genes and their consequences in metabolic syndrome. They observed that ADIPOQ, LPL, and PGC1α gene polymorphisms were found to be statistically different in the genotype comparison analysis. Furthermore, they showed the ADIPOQ + 45GG and P+P+ of LPL PvuII carriers had an increased risk of MS. Addionally, they observed that G allele of not only of LPL S447X but also PGC1α 482Ser allele were expected as protective factors. Very interestingly, all these SNPs were associated with body mass index (BMI), systolic blood pressure (SBP), serum high-density lipoprotein cholesterol (HDL-C), triglyceride (TG) and fasting blood glucose (FBG), adipokines and insulin as well as insulin resistance.
In summary, this is an interesting study with great potential. The entire manuscript is properly written with sufficiently documented, easily to follow, and well explained. The statistical methods used for data analysis are valid and correctly applied. Also, the results obtained in the study are well explained and clearly presented with appropriate number of tables of sufficient quality. However, there are some minor issues that should be corrected before the manuscript is suitable for publication.
Strengths:
Clinical relevance to patients with metabolic syndrome. The study is relevant in the field.
Weaknesses:
The manuscript is lacking some information that would allow authors to explain and described in the manuscript.
Minor comments:
- Abstract: Very poor way to present the background. They did not provide any rationale as to why they want to study polymorphisms of only these six gene? The organization of the abstract is very poor. Not enough background information is provided. The rationale is unclear. The abstract should be concise and crispy. The abstract needs to be re-written properly in a story flow.
- Introduction: In the introduction section need the recent literature about incidence of metabolic syndrome globally.
- In the introduction section, background information is not adequate. It is not easy to follow the motivation of this study. To filling this gap of information, need some literature about how different strategies are used to treat metabolic syndrome.
- The manuscript is poorly written, and requires a significant attention to improve punctuations, grammar and the readability. Spacing of the words need to be thoroughly checked throughout the manuscript. There are many places, authors should correct the typo errors. The manuscript needs an English language editing by a professional/company service to better reading.
- These kinds of studies have limitations. Hence, the author should have highlighted the potential limitations and shortcomings and suggests what could be the next step in this area of research.
Author Response
Dear reviewer,
Thank you for your review. All revisions we made were highlighted by “track changes” in the manuscript.
According to your recommendation, we made the following revision;
- The abstract was revised and fixed. We rewrote what was our motivation to study these gene polymorphisms in the abstract section. line 19-23
- The latest information on the Incidence/distribution of metabolic syndrome was added and cited in a recent paper to the Introduction section. lines 40 and 41.
- We revised the introduction section again and added contents about how different strategies are used to treat metabolic syndrome recently. line 105-112
- The main body text of the manuscript was revised, and the English language was edited to improve
- We highlighted the limitations one by one and made a somewhat discussion about the prospects of the field in the discussion section. line 446-447.
We welcome any comments. Thank you.